# Polyurethane Adhesives Based on Oxyalkylated Kraft Lignin

**DOI:** 10.3390/polym14235305

**Published:** 2022-12-05

**Authors:** Fernanda Rosa Vieira, Nuno Gama, Sandra Magina, Ana Barros-Timmons, Dmitry V. Evtuguin, Paula C. O. R. Pinto

**Affiliations:** 1Department of Chemistry, CICECO-Institute of Materials, Campus de Santiago, University of Aveiro, 3810-193 Aveiro, Portugal; 2RAIZ, Forest and Paper Research Institute, Quinta de S. Francisco, 3801-501 Aveiro, Portugal

**Keywords:** lignin-based polyol, propylene carbonate, oxyalkylation, polyurethane, wood adhesive, adhesion

## Abstract

Lignin-based polyol was obtained via oxyalkylation reaction with propylene carbonate using eucalyptus kraft lignin isolated from the industrial cooking liquor by the Lignoboost^®^ procedure. This lignin-based polyol (LBP) was used without purification in the preparation of polyurethane (PU) adhesives combined with polymeric 4,4′-methylenediphenyl diisocyanate (pMDI). A series of adhesives were obtained by varying the NCO/OH ratio of PU counterparts (pMDI and LBPs) and their performance was evaluated by gluing wood pieces under predefined conditions. The adhesion properties of the novel PU adhesive were compared with those of a commercial PU adhesive (CPA). The occurrence and extent of curing reactions and changes in the polymeric network of PA were monitored by Fourier transform infrared spectroscopy (FTIR) and dynamic mechanical analysis. Although the lap shear strength and glass transition temperature of the lignin-based PU adhesives have increased steadily with the NCO/OH ratio ranging from 1.1–2.2, chemical aging resistance can be compromised when the NCO/OH is very low. It was found that the lignin-based PU adhesive with an NCO/OH ratio of 1.3 showed better chemical resistance and adhesion efficiency than CPA possibly because the NCO/OH in the latter is too high as revealed by FTIR spectroscopy. Despite some lower thermal stability and shorter gelation time of lignin-based PU than CPA, the former revealed great potential to reduce the use of petroleum-derived polyols and isocyanates with potential application in the furniture industry as wood bonding adhesive.

## 1. Introduction

Rising concerns in our society regarding global warming have led to a growing demand for eco-friendly materials and components in the furniture sector to which wood-based composite belong. Adhesives are considered to be part of wood composites, thermosetting adhesives based on formaldehyde (i.e., urea, melamine urea, and phenol formaldehyde) are the most used. This is mainly driven by its great adhesive strength and price. However, they are very sensitive to hydrolysis, which can negatively affect formaldehyde emissions [1,2]. Furthermore, formaldehyde is considered a hazardous volatile organic compound (VOC) and in 2016 the European Chemicals Agency (ECHA) classified it as a carcinogen category 1B compound. Hence, the European stringent legislative requirements are forcing the adhesives producers to reduce the amount of formaldehyde or replace it by formaldehyde-free adhesives [3]. 

Polyurethane (PU) adhesives are formaldehyde-free adhesives suitable for application in wood composites, and present certain advantages over formaldehyde-based adhesives such as: resistance to weathering (moisture, temperature, UV radiation), feasible curing time, and flexibility in the formulation according to the final application [4,5]. PU adhesives are synthetized by the polyaddition reaction between a polyol and a polyisocyanate though other additives such as catalysts, flame retardants, and solvents are normally used as well [6]. However, the main reagents in the formulation, polyol and polyisocyanate, are derived from petroleum resources and their use and production are associated with climate change concerns. Despite the fact that the use of bio-based polyol in the production of PU is already a reality, especially those derived from vegetable oils, this is also a cause for concern due to competition with food supply. 

In turn, lignin, a natural aromatic polymer in most plant biomass sources containing hydroxyl groups, has been considered a potential renewable resource to replace polyol in PU formulations [7,8]. The major available source of lignin comes from the pulp and paper industry, where the lignin dissolved in the black liquor of kraft cooking and can be isolated by recently developed processes such as LignoBoost™ and LignoForce^™^ [9,10]. Being a natural polyol, lignin can be incorporated directly in the polymer matrix as filler or macromonomer in PU synthesis. However, due to its structural features, such as steric hindrance of part of the hydroxyl groups, mainly phenolic groups, lignin usually presents low reactivity with isocyanates, incompatibility and insolubility with other polymeric formulations, leading to aggregation which compromises the quality of the product (e.g., brittle and low strength product) [11,12,13,14]. Nevertheless, chemical modifications such as oxyalkylation and esterification to liquefy and increase the reactivity of the lignin-based polyol have been considered. It is noteworthy that esterification leads to a polyester polyol, while oxyalkylation leads to a polyether polyol.

The most studied routes to produce lignin-based polyether polyols (LBP) is oxyalkylation with propylene oxide (PO) [15,16,17]. However, in addition to health concerns, PO has a low boiling point and high vapor pressure, being a flammable compound, which requires special equipment to ensure safe conditions for carrying out this exothermic explosive reaction [18]. An alternative to the conventional oxyalkylation is the use of cyclic carbonates such as propylene carbonate (PC) and ethylene carbonate (EC). These compounds present high boiling points and low vapor pressure, being a safer alternative for the oxyalkylation of lignin [19]. This process has been evaluated and used to produce PU [20,21,22]. In fact, our group has optimized the oxyalkylation process to produce LBP with a suitable hydroxyl number (I_OH_) and viscosity for rigid foams and adhesive formulations [23]. 

In the production of lignin-based PU adhesives, most of the studies have explored the use of unmodified lignin in the formulation. However, as mentioned before, to improve the solubility of lignin with the conventional polyol or polyisocyanate, the use of solvents such as tetrahydrofuran (THF) or of a chain extender polyol such as poly(ethylene glycol) is required [24,25,26]. The use of lignin-based polyether polyol for PU adhesive is still little explored in the literature. Yet, Gouveia et al. [27] prepared lignin-based PU adhesives from kraft lignin oxyalkylated with PO under different conditions. In a similar manner, the same group prepared lignin-based PU adhesives by mixing castor oil with up to 30 % of LBP oxyalkylated with PC [28]. So far, to the best of our knowledge, these are the only studies that reported the use of oxyalkylated lignin with cyclic carbonate to produce PU adhesives.

Following our interest to develop bio-based materials, in this work, crude LBP obtained from oxyalkylation with PC was used as a substitute of petroleum-based polyols in the formulation of PU adhesives for wood bonding to be used in the furniture industry. The underlying hypothesis of this study was that the isocyanate groups of pMDI might easily react with both aliphatic OH groups of oxyalkylated lignin and the OH groups of wood substrates to afford a high crosslinking density, which in combination with the aromatic nature of lignin, would contribute to higher bonding strength and better chemical resistance of these adhesives than those obtained from petroleum-derived polyols. Indeed, due to the combination of these aspects it was also hypothesized that the amount of polymeric isocyanate required might be reduced. To verify this hypothesis, the NCO/OH ratio, referred to as the equivalence ratio of the polyisocyanates to polyols, was optimized by studying its effect on the properties of lignin-based adhesives. In addition, the wood-bonding properties, glass transition temperature (*T_g_*), as well as chemical and thermal resistance were evaluated and compared with those of a commercial PU adhesive.

## 2. Materials and Methods

### 2.1. Materials

The production and characterization of crude lignin-based polyol (LBP) was carried out according to our previous work, under optimized conditions using kraft lignin (KL) as the substrate, 1,8-diazabicyclo [5.4.0] undec-7-ene (DBU) as the catalyst was supplied by Aldrich Chemical Company (St. Louis, MO, USA) and propylene carbonate (PC) as oxyalkylation reagent was purchased from Acros Organics Company (Geel, Belgium) [23,29]. The final LBP consisted of PC-modified lignin and PC-derived oligomer (homo-oligomer) herein referred to as PC oligomer. Lignin-based adhesives were produced by reacting crude LBP with pMDI. The Voranate M229 pMDI was kindly supplied by Dow Chemicals (Estarreja, Portugal). In Table 1, the characteristics of crude LBP and pMDI are presented. The CPA is a polyurethane pre-polymer with a gelation time of minutes.

### 2.2. Preparation of Lignin-Based PU Adhesives

Lignin-based PU adhesives were prepared in a polypropylene cup under nitrogen atmosphere, where the pMDI was slowly added to the crude LBP. The mixtures were stirred with rotating blades, for about 30 s at 1000 rpm. Five formulations using different NCO/OH ratios (1.1, 1.3, 1.6, 1.9 and 2.2) were prepared, being the required content of pMDI calculated according to the Equation (1). Note that the amount of water present in the polyols was considered in the calculation to ensure the appropriate NCO/OH ratio.
(1)$$R_{NCO/OH}=\left(m_{iso}\times {\%}_{NCO}/M_{NCO}\right)/\left(m_{polyol}\times \left(I_{OH}\right)/M_{KOH}+\left(m_{H2O}\;\right)\times Eq_{H_{2}O}\;\right)$$
where *R_NCO/OH_* is defined as the number of moles of NCO groups of the isocyanate per OH moles of polyol and water, *m_iso_* is the mass (g) of isocyanate, *%_NCO_* is the quantity of NCO groups in the isocyanate (31.1%) and *M_NCO_* is the molecular weight of the NCO group. *m_polyol_* is the mass (g) of each polyol, *I_OH_* is the hydroxyl number of polyol (mg KOH/g). *M_KOH_* is the molecular weight of KOH. *m_H_*_2*O*_ is the mass of water present in the polyol. *Eq_H_*_2*O*_ is the equivalent of OH groups present in the water. 

The obtained lignin-based PU adhesives were cured during 24 h at room temperature prior to characterization and the lap shear tests. A commercial PU adhesive (CPA) was used to compare the results with lignin-based PU adhesives.

### 2.3. Characterization of PU Adhesives

The PU adhesives were characterized by the Fourier transform infrared spectroscopy (FTIR) after being cured at room temperature during 24 h. The spectra were recorded on an FTIR System Spectrum BX (PerkinElmer) coupled with a universal ATR sampling accessory, 4000 to 500 cm^−1^ by accumulating 64 scans with a resolution of 4 cm^−1^.

For the lap shear tests, an Instron 5966 universal mechanical test analyzer was used. The measurements of the adhesive strength of adhesives in the wood-to-wood specimens were carried out according to ASTM D 906 standard [30], with a crosshead speed of 10 mm.min^−1^. The values presented correspond to the average of five specimens. Strips of the pine wood with a flat and cleaned surface were cut into specimens (25 × 100 × 3 mm^3^), and then pairs of strips were glued using the lignin-based PU adhesives with a contact area of 25 × 30 mm^2^. A grip was used to keep the jointed wood strips at constant external pressure for 24 h at room temperature for adhesive curing prior to lap shear tests.

Green strength is an important property of adhesives, which shows the ability of an adhesive to hold substrates together when brought into contact and before the adhesive develops ultimate bond properties when fully cured [31]. Hence, the green strength is measured over a period of time to determine when the adhesive is fully cured, which corresponds to the moment when the values of lap shear strength reach a constant value. For the determination of green strength time, a PU adhesive produced using an NCO/OH ratio of 1.1 was selected. Bonded wood specimens were submitted to the lap shear test at different curing times.

The *T_g_* values of PU adhesives were determined using the material pocket accessory by Dynamic mechanical analyses (DMA). The analyses were carried out using Tritec 2000 equipment (Triton Technologies, Leicestershire, UK), from −50 °C up to 100 °C at a constant heating rate of 2 °C.min^−1^ at a frequency of 1 Hz and 10 Hz.

The thermogravimetric analysis (TGA) was performed to evaluate the thermal stability of the PU adhesives using a SET- SYS Evolution 1750 thermogravimetric analyzer (Setaram, Caluire, France) from room temperature up to 800 °C, at a heating rate of 10 °C/min and under oxygen flux (200 mL/ min).

The chemical resistance of the commercial and lignin-based PU adhesives (produced using NCO/OH ratios of 1.1 and 1.3) was evaluated under different conditions before being submitted to the lap shear tests. Each group of five jointed wood strips specimens was cured for 3 days prior to being immersed in the different environments: cold water (30 °C) for 1 day; hot water at 100 °C for 1 h; acidic water (pH = 2) at 80 °C for 1 h; and alkaline water (pH = 10) at 80 °C for 1 h. After the treatments, the samples were washed with distilled water, dried at room temperature, and submitted to the lap shear tests.

## 3. Results and Discussion

Wood adhesives of satisfactory bonding quality should comply with many requirements such as curing timing, cohesion/adhesion strength, durability, consistency, wetting, and temperature [4]. For example, durability is related to the environmental conditions that affect the aging of the adhesive (moisture, heating, cooling, chemical action, light exposure, etc.) to which the product will be exposed. Accordingly, in this study, not only the main mechanical and bonding properties of lignin-based PU adhesives in wood bonding essays were evaluated, but also the effect of chemical ageing of the applied adhesive. The lignin-based polyol (LBP) used for the adhesive syntheses was obtained by oxyalkylation of lignin isolated from the black liquor of eucalyptus wood under optimized conditions. The hydroxyl number (I_OH_) of LBP was 225 mg KOH/g and its viscosity (0.56 Pa^.^ s) allowed effective mixing with diisocyanate (pMDI) to prepare different adhesives formulations [23].

### 3.1. Gelation Time and Green Strength

The gelation time or pot life is the maximum time that the adhesive is in a fluid state suitable to be applied onto the substrate. In general, the lignin-based adhesives proved to be more reactive than the conventional PU adhesive (CPA). In fact, as presented in Table 2, its gelation time is around 50% shorter compared to CPA, which can be attributed to the residual DBU catalyst present in the crude LBP, which is known to be widely used in the PU formulations as catalyst. With regard to the gelation time of lignin-based PU adhesives, it decreased slightly as the NCO/OH ratio increased in the formulation, though this tendency is not clear as the values are close and very short. The same behavior was observed by Sahoo et al. [32], who reported that the gel time of the bio-based PU adhesive decreased as the NCO/OH ratio increased from 1.1 to 1.5. This is due to the higher concentration of highly reactive NCO groups, which besides reacting with the OH groups of the polyol, those in excess can also react with the OH groups of wood or even the water of wood (humidity) forming covalent bonds with the wood [33]. Nevertheless, this effect would equally affect both CPA and lignin-based PU adhesives.

Another important property of adhesives is their green strength, which is the strength measured before the cure is completed. This property indicates the early development of bond strength, before achieving the ultimate strength [31]. To assess the curing performance of lignin-based adhesives only the PU 1.1 (NCO/OH = 1.1) was considered, since it is the formulation whose gelation time was longer; thus, the remaining ones would necessarily react faster. For that purpose, the lap shear strength was monitored for 5 days, and the results are shown in Figure 1. As can be seen, that lap shear strength of PU 1.1 increased slightly over time: from the 1st to the 2nd day a 7% increase was registered; from 3rd to 4th day the increase was circa 12.5%; and after the 4th day, the lap shear strength tends to go back down and stabilizes. These variations might be due to the ongoing cross-linking reactions between the excess of NCO groups and OH groups of water from humidity or even the OH groups of the wood specimen; yet, they fall within the error of the measurements. These results indicate that, in fact, the lignin adhesive was practically cured from day one. Therefore, green strength values have probably been reached before 1 day.

### 3.2. Effect of NCO/OH Ratio on the Adhesion Efficiency and Properties of Lignin-Based PU Adhesives

The bonding properties of PU adhesives as well as their chemical, physical, and mechanical performance are strongly influenced by the NCO/OH ratio. Generally, in PU adhesive formulations, NCO/OH ratios higher than one are used to ensure the formation of urethane bonds and to increase the adhesive bonding strength [34] as the NCO groups in excess can react with OH groups of wood forming covalent bonds with the subtract [33]. In that sense, to monitor the formation of urethane bonds formed during the polymerization reaction between the NCO groups of pMDI and OH groups of crude LBP, FTIR analysis was used. In Figure 2, the FTIR spectra of crude LBP, CPA and of lignin-based PU adhesives (PU 1.1, 1.3, 1.6,1.9, and 2.2) are presented. As can be seen, all lignin-based PU adhesive present similar profiles. They are mainly characterized by the stretching vibrations of the C=O groups at 1780 cm^−1^, which is associated with the linear carbonate linkages of crude LBP formed by the reaction of the propylene carbonate with the aliphatic OH of lignin [29]. The bands around 1700 cm^−1^ and 1090 cm^−1^ correspond, respectively, to the stretching vibration of the C=O and C–O linkages of the urethane moieties are observed in all PU adhesives [35]. The other characteristic bands of PU, observed at 2860 cm^−1^ and 2960 cm^–1^ are associated with the alkane C–H stretching vibration, while the N–H stretching vibration is observed in the range of 3360–3250 cm^−1^. The main difference in the spectra that can be observed is the typical free isocyanate (NCO) stretching band at 2270 cm^−1^. As expected, the intensity of this band increased with the increase of NCO/OH ratio. In addition, the band at 2270 cm^−1^ of the PU 2.2 presents similar intensity when compared with that of the CPA which may indicate that the latter was prepared using a similar NCO/OH ratio. 

The adhesion efficiency of lignin-based PUs was measured using the lap shear test after 3 days of curing to ensure that the adhesives were fully cured. The results presented in Figure 3 show that the average lap shear strength of lignin-based PU increased as the NCO/OH ratio increased up to 2.2. This is attributed to the reaction of free NCO with OH groups of wood and wood-adsorbed water, which increases the crosslinking density of PU network [36,37]. In addition, the free NCO groups also form a variety of complementary linkages, increasing the strength of PU adhesives [26,38]. From these results, it can also be observed that the CPA (which according to the FTIR spectrum was formulated using an NCO/OH ratio around 2.2) presented a lap shear strength value of 3.1 MPa. This bonding performance is similar to that of the LBP-based adhesive synthetized using the NCO/OH ratio of 1.1 confirming that the lignin-based polyol yields adhesives with superior bonding performance using smaller amounts of isocyanate. In fact, when comparing adhesives prepared using similar NCO/OH ratios, one can see that the lap shear strength of lignin-based PU (PU 2.2) is 70% higher than that of the CPA counterpart.

Besides the lap shear strength, the type of failure mode was also used to evaluate the adhesive performance. Table 3 and Figure 4 summarize the observations made and as it can be observed the failure modes for the lignin-based PUs include cohesive failure (CF), adhesive failure (AF) and substrate failure (SF) but no specific correlation could be discerned between the NCO/OH ratio and the mode of failure. Nonetheless, for the adhesive prepared using the lowest NCO/OH ratio (1.1), the failure mode is mainly CF (Figure 4), which can be attributed to the less crosslinked structure of the bulk PU adhesive and better wettability of the wood surface with either LBP or PC oligomers, which confers stronger hydrogen bonding with the OH groups of the wood surface. As a result of this combination of factors, failure occurs within the adhesive material itself, just as observed for CPA (Table 3). As the NCO/OH ratio increases other failure modes (CF, and SF) are also observed. This is attributed to the excess of NCO, which leads to increased stiffness of PU and embrittlement of the adhesive layer of the lap joint, as well as the higher number of covalent bonds formed between the isocyanate and the OH groups of the wood substrate [39]. The superior performance of lignin-based adhesives compared to CPA can be attributed to the following reasons: (1) the aromatic nature of lignin, which contributes to their stiffness [40,41], and (2) the modification of lignin, which yields a polyether with mainly primary OH groups which react promptly with the polyisocyanate. Thus, increasing the crosslinking density of the polymer matrix and subsequently the adhesive strength [28]. Furthermore, the crude LBP contains residual catalyst and PC-oligomer, which may promote the crosslinking of the polymeric network, as well as enhance the wettability of the surface and convey some flexibility [25]. The mode of failure observed for NCO/OH = 1.6 (SF) (Table 3) suggests that this formulation did not spread well on the wood surface of the lap joint because its stiffness should not be higher than those of the samples prepared using higher NCO/OH ratios. In fact, the lap shear strength of that sample has an intermediate value. 

DMA using a pocket accessory was used to determine the *T_g_* of cured PU adhesives, measured from the top of tan δ. As can be seen from Figure 5, the *T_g_* of lignin-based adhesives increases as the NCO/OH ratio increases, from 16.5 °C (PU 1.1) up to 83 °C (PU 2.2). This is due to the higher crosslinking density derived from the higher amount of NCO and the presence of residual catalyst in the crude LBP, which limits the polymer chain movements thus reducing the flexibility. Similar behavior was also observed for PU systems using bio-based polyols using different NCO/OH ratios [32,36]. In turn, the *T_g_* of CPA is around 50 °C, indicating that this is the maximum operating temperature range of this adhesive. However, it was possible to conclude from the FTIR analysis that this commercial PU adhesive was produced using a higher NCO/OH ratio. Even though the *T_g_* of PU 1.3 is around 10 °C lower than that of CPA, the results indicate that it is possible to use a lower amount of isocyanate in the formulation of PU adhesives using crude LBP to obtain lignin-based adhesive with a comparable range of operating temperature, which is a huge advantage since isocyanate can cause health concerns for workers and consumers with occupational asthma [42,43].

### 3.3. Thermal and Chemical Resistance of Lignin-Based PU Adhesives Formulated with 1.1 and 1.3 NCO/OH Ratio

Although the lignin-based adhesives produced using the highest NCO/OH ratio, PU 2.2, presented the highest lap shear strength, PU 1.1 and PU 1.3 were the formulations that yielded values of lap shear strength and of *T_g_* closer to those of the CPA, making them potential substitutes of the CPA. Thus, their thermal resistance (Figure 6) and chemical resistance (Figure 7) were evaluated and compared with those of CPA.

In general, PUs presented three main degradation steps: i) below 200 °C attributed to the unreacted isocyanates, the labile moieties of LBP and PC oligomers, (ii) between 200 and 300 °C assigned to the thermal degradation of urethane moieties, and iii) above 400 °C related to the thermal decomposition of the ether linkages of the polyol segments (Figure 6a) [44,45,46]. From the results shown in Figure 6b, it can also be observed that increasing the NCO/OH ratio results in the reduction of the rate of mass loss. The maximum degradation rate of first mass loss of PU 1.1 occurs at around 160 °C, while that of PU 1.3 at 190 °C. This is due to the higher crosslinking of the PU adhesive, which contains fewer thermally labile free polyols (LBP and PC oligomers). In turn, the maximum rate of degradation of the first mass loss of CPA takes place at 278 °C. The difference between the thermal degradation of bio-based adhesives and CPA is due to the more thermally labile LBP and PC oligomers compared to the petroleum-derived polyols used in the production of CPA. Finally, probably due to the aromatic nature of LBP, the PU adhesive (e.g., PU 1.3) showed a higher temperature of network degradation leading to char formation (495 vs. 488 °C) when compared to CPA. 

With regard to the results obtained for the chemical resistance tests depicted in Figure 7, generally the lignin-based adhesives showed good chemical resistance to cold water treatment, especially the formulation with NCO/OH ratio = 1.3. However, its adhesion performance was substantially reduced in the hot water assay and during ageing under acidic and alkali conditions. This loss in the adhesion strength under alkaline conditions was especially noticeable for the PU 1.1 formulation. Most of other studies reported in the literature revealed the same behavior when acid and alkali conditions were used as they lead to the extensive hydrolysis of the urethane and ether/ester bonds, depending on the type of polyol used [31,36,39,47]. Noteworthy that when the crude LBP reacted with pMDI at NCO/OH ratio of 1.3 revealed a greater chemical resistance than CPA. This finding can be attributed to the alkali and acid resistance of modified kraft lignin, which makes the respective PU adhesive more resistant to chemical ageing.

## 4. Conclusions

In this study, lignin-based adhesives for wood bonding were synthesized using pMDI and crude LBP obtained from oxyalkylation of KL with PC and compared with a commercial PU adhesive (CPA). The effect of different NCO/OH ratios (1.1 to 2.2) on the adhesion properties of PU adhesives was evaluated and related with gelation time, general structure, thermal behavior, wood bonding strength and chemical resistance. All LBP-based adhesive formulations showed lower a NCO/OH ratio than CPA as revealed from FTIR analysis. Among the lignin-based adhesives examined, the formulation with an NCO/OH ratio of 1.3 (PU 1.3) had one of the longest gelation times (3 min). The value of *T_g_* (40 °C) and lap shear strength (3.1 MPa) are close to the values of CPA, and the chemical resistance proved to be superior to that of CPA by around 10% higher for acid and cold-water conditions. In turn, even though the PU 1.3 adhesive showed shorter gelation time and lower thermal stability than CPA, it presented a slower degradation rate due to the aromatic nature of crude LBP. Furthermore, using LBP, it was possible to produce an adhesive using less isocyanate than that found in the commercial PU adhesive, which brings technical and environmental advantages. This study clearly demonstrates that lignin-derived PU adhesives can be used as an alternative to commercial adhesives using petroleum-derived polyols.

## Figures and Tables

**Figure 1 polymers-14-05305-f001:**
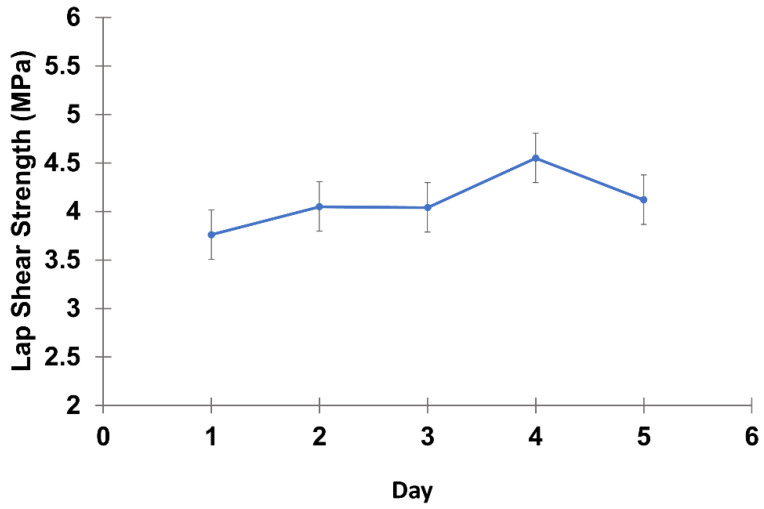
Lap shear test versus curing exposure period of lignin-based PU adhesive formulated with NCO/OH = 1.1 (PU 1.1).

**Figure 2 polymers-14-05305-f002:**
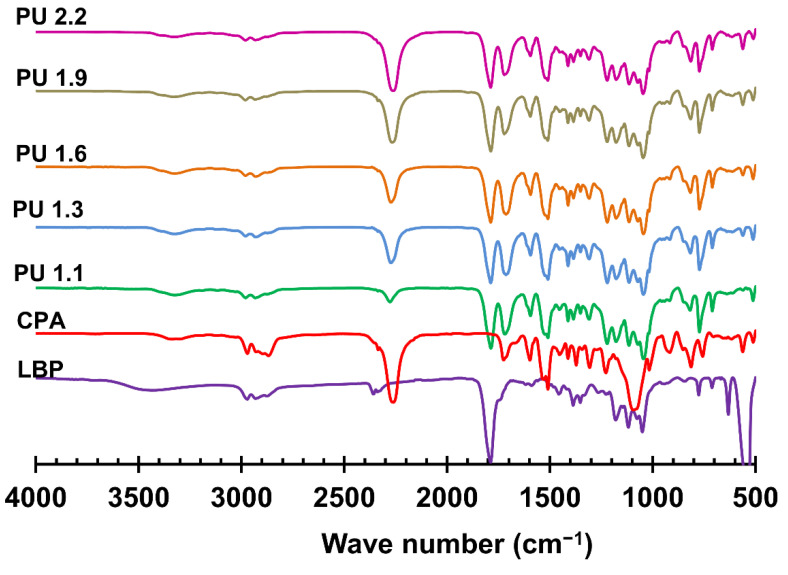
Normalized FTIR spectra of crude LBP, CPA, and lignin−based PU adhesives with different NCO/OH ratios (PU 1.1,1.3, 1.6, and 2.2).

**Figure 3 polymers-14-05305-f003:**
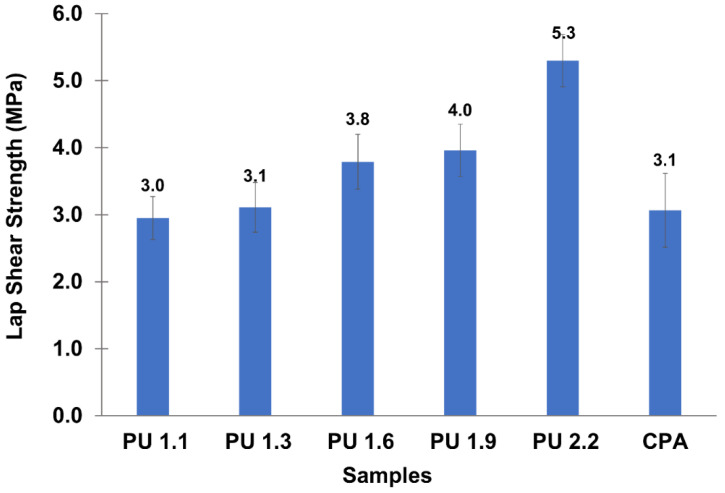
Effect of NCO/OH ratio on average lap shear strength of the lignin-based PU adhesives compared with CPA.

**Figure 4 polymers-14-05305-f004:**
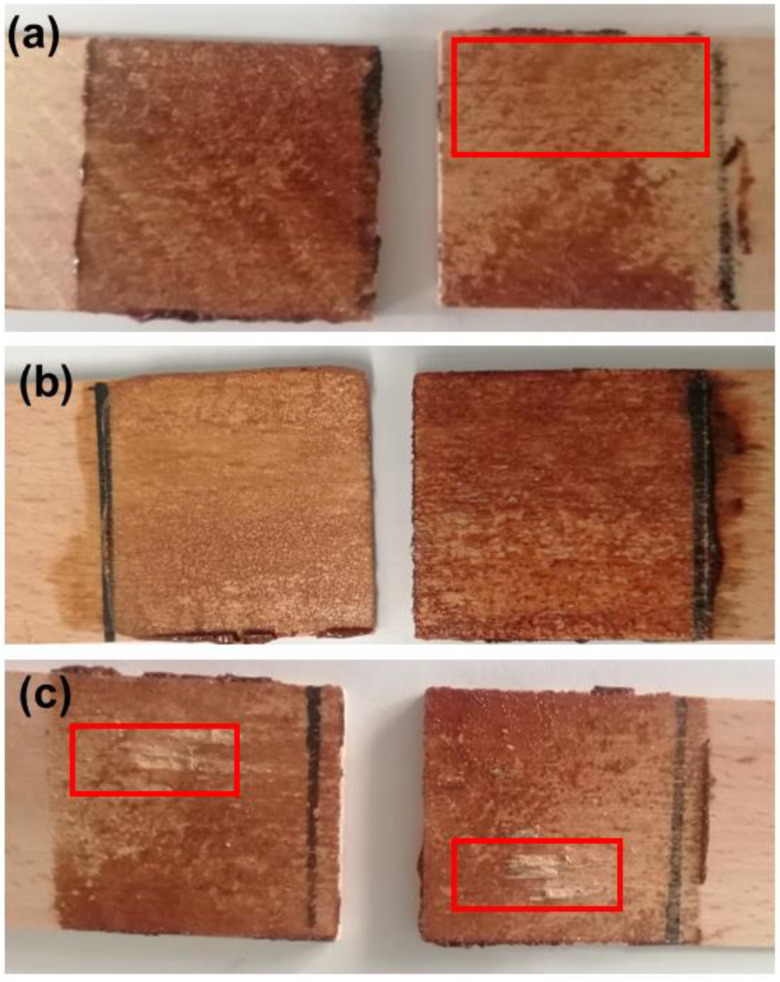
Examples of (**a**) adhesive failure (AF), (**b**) cohesive failure (CF), and (**c**) substrate failure (SF).

**Figure 5 polymers-14-05305-f005:**
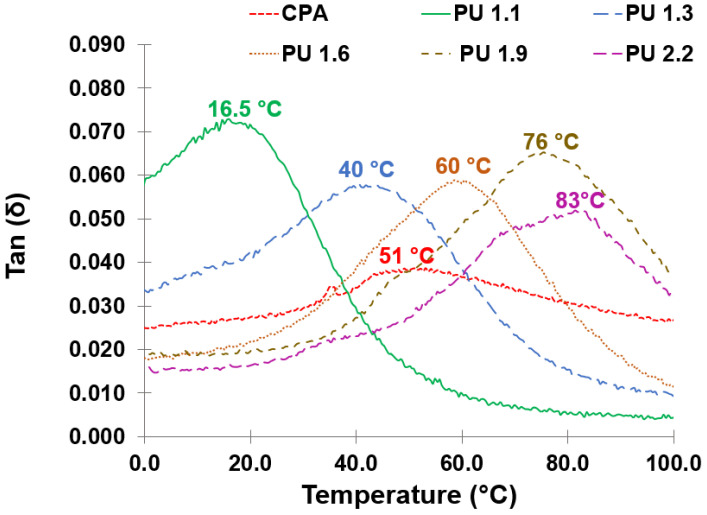
DMA results of lignin-based PU adhesives with different NCO/OH ratios (PU1.1 to PU 2.2) and CPA obtained at frequency 1 Hz.

**Figure 6 polymers-14-05305-f006:**
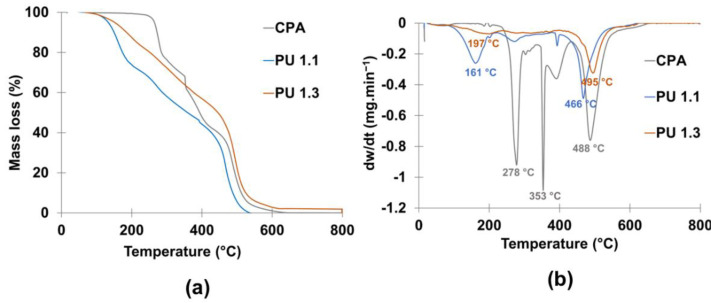
(**a**) TGA curves of PU 1.1, PU 1.3, and CPA, (**b**) Derivatives of mass loss (dw/dt) of PU 1.1, PU 1.3, and CPA.

**Figure 7 polymers-14-05305-f007:**
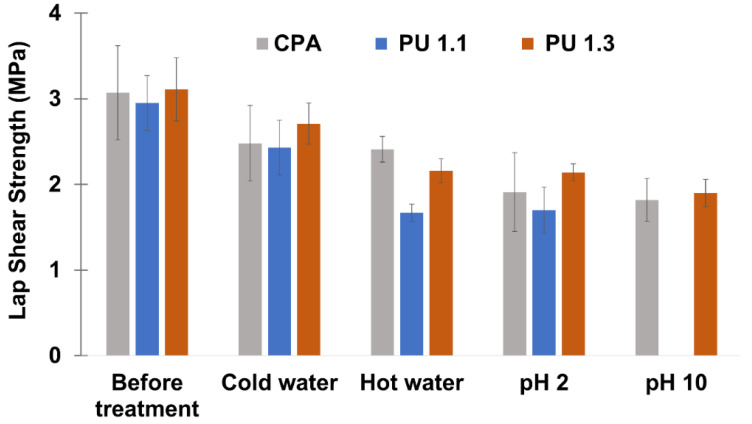
Chemical resistance of PU 1.1, 1.3, and CPA.

**Table 1 polymers-14-05305-t001:** Crude LBP and pMDI characteristics.

Characteristics	Crude LBP	pMDI
Type	Aromatic polyether	Aromatic diisocyanate
I_OH_, mg KOH/g	225	-
NCO content, %	-	31.1
Water content, %	0.39	-
Viscosity, Pa. s	0.56	0.19
Mw, Da	1700	-
PC-oligomer, %	15.5	-

**Table 2 polymers-14-05305-t002:** PU adhesives formulations and their gelation time.

Samples Codes	Crude LBP * (pbw)	pMDI * (pbw)	NCO/OH Ratio	Gelation Time (min)
PU1.1	100	59.4	1.1	3.0
PU 1.3	100	70.1	1.3	3.0
PU 1.6	100	86.3	1.6	2.4
PU 1.9	100	102.5	1.9	2.4
PU 2.2	100	118.7	2.2	2.1
CPA	-	-	-	>5 **

* Part by weight; ** confidentially reasons the exact value is not provided.

**Table 3 polymers-14-05305-t003:** The effect of NCO/OH ratio on the mode of failure.

Samples Codes	NCO/OH Ratio	Mode of Failure
PU1.1	1.1	CF
PU 1.3	1.3	CF + AF + SF
PU 1.6	1.6	SF
PU 1.9	1.9	CF + AF + SF
PU 2.2	2.2	CF + AF + SF
CPA	-	CF

## Data Availability

Data sharing not applicable.

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
