# Peer review of "Polyurethane Adhesives Based on Oxyalkylated Kraft Lignin"

_polymers, 2022, doi:10.3390/polym14235305_

Round 1

Reviewer 1 Report

State potential applications in abstract and introduction

State hypothesis at the end of Introduction

Any statistical analysis in Methods?

Materials - state city and country from where the materials were obtained.

Figure 1 - what is the difference between green strength and lap shear strength? Use single term only

Take standards as references

it is suggested Characterization is subdivided into sections like mechanical, thermal etc

Acknowledgements - state country

Section 3 - Results and Discussion

Put numerical finding in Conclusions

Reviewer 2 Report

The presented work concerns the preparation and testing of the properties of polyurethane adhesives in which the diol component is a modified kraft lignin. The subject of the work is very interesting. The text is written in an accessible way, but requires a few explanations and supplements.

Abstract:

line 14: please correct 4,4'-methylene diphenyl dissocyanate to 4,4'-methylenediphenyl diisocyanate

Introduction:

The introduction is written correctly, all the information presented is related to the topic of the work, and the cited literature is correctly selected. The purpose of the work is also properly presented.

Materials and methods:

line 106: please correct PMDI on pMDI, additionally standardize pMDI throughout the text (cannot be pMDI).

Results:

Lines 225-230: vibration C=O

Please correct the description, it is incorrect. The vibrations at around 1780cm-1 are related to the C=O vibrations of the carbonate group, while the C=O vibrations of the urethane group are at around 1700cm-1. The FTIR CPA spectrum shows no vibration at 1780, while the LBP spectrum shows this high intensity band! Therefore, the description provided is incorrect. Additionally, please cite the following papers in which the FTIR spectra of polyurethanes are described in detail (including the ranges of C=O stretching vibrations):

doi: 10.3390/polym14142933

Line 232: band at the 3440cm-1 is not responsible for the vibrations of the N-H group in polyurethanes, only for the vibrations of the -OH group. In the adhesives obtained, this band is absent, it is only visible in the spectrum obtained for LBP, i.e. for polyol. Please correct it!!

Thermal analysis:

In my opinion, a better tool for determining and analyzing changes in the value of glass transition temperature would be the DSC. However, the DMA analysis is also acceptable and its results would have to be properly presented. Where were the conservative modulus and loss values obtained? What is the vibration damping? The authors of the methods write that DMA was performed at frequencies of 1 and 10 Hz. For what frequency are the results shown? Why has the activation energy for individual transformations not been determined? DMA analysis is a very interesting research technique that gives a lot of information about the material under study. It is a pity that the authors used it to a negligible extent. Please expand the DMA description with items that are missing.

Thermal stability

lines 322-325: at 300st C, ether bonds do not break, only urethane bonds, the ether bonds are decomposed above 400 degrees. Please correct it and quote relevant works, e.g.

doi:10.1002/pat.4083 ; doi:10.3390/polym10050537

Round 2

Reviewer 2 Report

The submitted text has been improved compared to its previous version. All my comments have been taken into account, I have no more comments. my recommendation: accept